# Therapy for Non-Diabetic Foot Infections After Orthopedic Foot and Ankle Surgery: The Duration of Post-Debridement Antibiotic Treatment Does Not Alter Failure Risk

**DOI:** 10.3390/life14111509

**Published:** 2024-11-20

**Authors:** Martina Scherer, Laura Soldevila-Boixader, İnci Yιldιz, Franziska C. S. Altorfer, Pascal R. Furrer, Silvan Beeler, Stephan H. Wirth, Arnd Viehöfer, İlker Uçkay

**Affiliations:** 1Department of Orthopedic Surgery, Balgrist University Hospital, 8008 Zurich, Switzerland; martina.scherer@balgrist.ch (M.S.); inci.yildiz@balgrist.ch (İ.Y.); franziska.altorfer@balgrist.ch (F.C.S.A.); pascal.furrer@balgrist.ch (P.R.F.); silvan.beeler@balgrist.ch (S.B.); stephan.wirth@balgrist.ch (S.H.W.); arnd.viehoefer@balgrist.ch (A.V.); 2Infectious Disease Service, IDIBELL-Hospital Universitari Bellvitge, Universitat de Barcelona, Feixa Llarga s/n, Hospitalet de Llobregat, 08907 Barcelona, Spain; laura.soldeb@gmail.com; 3Infectiology, Unit for Clinical and Applied Research, Balgrist University Hospital, 8008 Zurich, Switzerland

**Keywords:** adult foot and ankle surgery, infection, surgical site infections, surgical debridement, duration of antibiotic treatment, treatment failures, total ankle prostheses

## Abstract

Despite growing numbers of scientific publications on the optimal antibiotic treatment for diabetic foot infections, the data on the adult population with non-diabetic (postsurgical) foot infections is limited. Therefore, one of the largest single-center databases at the Balgrist University Hospital in Zurich, Switzerland, was established between January 2014 and August 2022. Using a case–control study design, we retrospectively investigated failures of combined surgical and antibiotic therapy for surgical site foot infections (SSIs). Overall, 17.4% of the episodes experienced therapeutic failures, particularly in patients with infected ankle prostheses. However, age, biological sex, pathogens, the duration of post-debridement antibiotic treatment, the number of surgical debridements, or the use of negative-pressure wound care altered the failure risk. In the multivariate logistic regression analyses, the duration of postsurgical antibiotic use was completely indifferent (as a continuous variable with an odds ratio of 1.0 and a 95% confidence interval ranging from 0.96 to 1.03) when stratified into inter-tertiary groups. Our findings suggest that shorter courses of systemic antibiotics may be appropriate in non-diabetic adults, supporting better antibiotic stewardship. Ongoing randomized controlled trials are under way to investigate which patients might safely receive shorter antibiotic treatments for surgical site infections following elective foot and ankle procedures.

## 1. Introduction

According to the available scientific literature, most surgical site infections [1] (SSIs) after elective foot and ankle surgery in healthy adults may have an incidence rate between 1.9% [2] and 4.2% [3]. In contrast, among adult patients with diabetes, this incidence rate can reach 9.1% or higher [4,5]. Classical risk factors for SSIs include patients’ co-morbidities (American Society of Anesthesiologists’ scores ranging from 3 to 5), revision surgeries, large metallic implants, long operation times [1], inadequate perioperative antibiotic prophylaxis, the use of allografts, bone substitutes, smoking, and postoperative wound problems [2,6,7,8]. For traumatic foot surgery, such as open fractures [9], this risk increases to up to 33% in Gustilo grade IIIc [9]. It is aggravated by a selection of multi-resistant Gram-negative pathogens by concomitant excess and the prolongation of prophylactic antibiotic use. Another factor that clinicians know of and that is difficult to assess in retrospective studies is patients’ non-compliance with wound care and off-loading.

The microbiology of foot infections can vary depending on the nosocomial origin, underlying implants and ulcerations, the geographical region, and associations with other pathologies such as ischemia or skin maceration. In acute, non-diabetic orthopedic SSIs in temperate climates such as Switzerland, the predominant pyogenic bacteria causing monomicrobial infections primarily stem from Gram-positive species, including staphylococci, streptococci, and enterococci. Skin commensals are usually encountered in implant-related infections. In patients with diabetic foot infections, particularly those presenting with existing or previous foot ulcerations or skin macerations, the microbiological profile demonstrates a distinct shift towards enterococci, Gram-negative pathogens, or polymicrobial infections involving skin commensals. However, in subtropical regions and areas with a higher proportion of multi-resistant (Gram-negative) germs, this epidemiology can differ substantially. As two examples, Meng et al. examined the incidence and risk factors for SSIs following elective foot and ankle surgery in China [7]. The microbiological culture results reveal a predominance of *Pseudomonas aeruginosa,* followed by methicillin-resistant *Staphylococcus aureus* (MRSA) and coagulase-negative staphylococci. In the Southern USA, Truong et al. evaluated 113 adult patients with foot infections between 2011 and 2019 and found that 83 patients had diabetes and 30 were non-diabetic. Polymicrobial infections were more common in patients with diabetes (83.1% vs. 53.3%; *p* = 0.001). The predominant pathogens were *S. aureus* in patients with diabetes and *P. aeruginosa* in those without diabetes. Anaerobes (9.4%) and fungal infections (3.1%) were not common in both groups [10].

Despite a global increase in orthopedic foot procedures in recent years, most of the literature on the antibiotic treatment of foot infections targets adult patients with diabetes and (implant-free) diabetic foot infections (DFIs) [11]. However, DFIs are a distinct clinical entity mostly lacking the hallmark of established polyneuropathy or progredient (terminal) ischemia, which heavily complicates their therapeutic approach. However, patients with DFIs usually lack implant-related infections [12]. Other considerations include the correction of hyperglycemia and the extent of surgical debridement versus minor amputations, which remain valuable options in diabetic foot osteomyelitis [13]. The lack of optimal compliance to off-loading is probably the sole commonality between non-diabetic patients with foot infections and those with classical DFIs. Moreover, the discrepancies between the two entities extend beyond epidemiological characteristics, encompassing therapeutic success as well. In DFIs, the overall clinical failure rate after treatment is high, and recurrence (albeit mostly with different germs) occurs in around one in four patients independent of the microbiological adequacy of the initial antibiotic regimen [11].

Our research center largely publishes studies on therapeutic failures and complications in DFIs [13,14,15]. The aims of this current study are different: we investigate the remission rate (or inversely, failure) of SSI therapies in adult elective foot and ankle surgery. We renounce the investigation of specific risk associations with SSIs in foot surgery, which has been discussed in a prior article [2]. Importantly, we hypothesize that the duration of postsurgical antibiotic treatment and the number of surgical debridement in the operating theater does not influence the risk of failure, leaving room for better antibiotic and surgical stewardship [16] in these infection patients. Ultimately, our retrospective results enable us to tailor future prospective randomized studies to orthopedic foot infections [17].

## 2. Materials and Methods

A retrospective analysis on all SSIs after elective foot and ankle surgeries was conducted at our tertiary foot center, an orthopedic referral center at the Balgrist University Hospital in Zurich, Switzerland, between January 2014 and August 2022 [2]. We used data from the hospital’s medical databases and confirmed the presence of SSI by analyzing each electronic file. We included all SSIs (first-time episodes) after adult elective foot and ankle surgery in patients older than 18 years with a minimum surveillance period of two years after treatment start and with detailed information available at database closure (31 August 2024). The definition of an SSI was made according to the international criteria [1,2].

The minimal required variables for each infection episode were information on index surgery; type of surgical and antibiotic therapies during follow-up; information on the biological sex, age, and body mass index (BMI); whether the patient has diabetes mellitus; pathogens (especially *S. aureus* yes or no); the presence of total ankle prosthesis (TAA); immobilization (off-loading); the duration of systemic antibiotic prescription in days; the number of surgical debridements for infection; and the use of negative-pressure wound devices [18] during hospitalization or outpatient setting. Exclusion criteria have previously been described in different studies [2,19,20] and briefly resumed as recurrent infection episodes; amputations without residual infection; infection episodes with prior emergency index surgery; external patients with partial treatment; open fractures [9]; foot infections exceeding the ankle (e.g., gas gangrene, necrotizing fasciitis, and other rapidly spreading severe soft tissue infections); insufficient documentation; antibiotic therapy relying mostly on local (topical or intraosseous) antibiotic therapies [2,14]; atypical pathogens such as *Actinomyces* spp., fungi, or mycobacteria; concomitant severe and remote infections such as endocarditis or brain abscesses; and, most importantly, the presence of a community-acquired DFI with ischemia, chronic infected foot ulcers, and/or severe polyneuropathy. In contrast, a patient with well-regulated concomitant diabetes mellitus was included in the study if they had developed an SSI (e.g., a patient with diabetes with a non-ischemic SSI of their orthopedic hardware in the foot). The accuracy of glycemic control was based on clinical judgment by the consulting internal medicine department.

### 2.1. Study Objectives and Definitions

We performed a case–control study with two primary outcomes: remission after minimal follow-up and Microbiological Recurrences. The latter was defined as a recurrent infection at the same anatomical localization, after terminating the antibiotic therapy, showing the same pathogen(s) as in the index infection episode. A new infection with a new pathogen or at a different anatomical localization (e.g., in the other foot) was considered a “Clinical Failure”. The differentiation between “Clinical Remission” and “Microbiological Recurrence” was based on our assumption that overall failure might be due to a panoply of explications, whereas Microbiological Recurrence reflects a failure of the antibiotic therapy per se. Remission was defined as a complete absence of any clinical, laboratory, and/or radiological evidence of a new or persistent problem. The use of serum laboratory controls during treatment for orthopedic infections is a widespread practice; however, firm scientific evidence is lacking [19,20]. Therefore, we refrained from using them routinely.

An SSI was defined based on international standards [1,21] with occurrence at the surgical site within 30 days postoperatively (one year for implant-related surgeries) [21]. Furthermore, in the case of clinical suspicion of infection (purulent wound discharge, erysipelas, lymphangitis, dolour, rubor, calor, and/or functio laesa), infection was confirmed by taking several deep tissue and bone samples (mostly intraoperative specimens) for microbiological and histological analyses and in the case of implants for sonication. Additional microbiological examinations, such as polymerase chain reaction, were ordered occasionally. The period of incubation of the intraoperative cultures was 14 days independent of prior antibiotic prescription. The Institute for Medical Microbiology at the University of Zurich processed the samples using VITEK 2 and/or routine cultures based on EUCAST criteria (https://www.eucast.org/clinical_breakpoints; accessed on 11 October 2024) for susceptibility testing. 

### 2.2. Statistical Analysis

The primary study objective was identifying treatment failures (or inversely remissions) after a combined surgical approach for SSI after orthopedic foot surgery. This was a case–control study. The cases were either “Clinical Failures” or “Microbiological Recurrences”, and the controls were “remissions”. We used descriptive statistics and compared groups with the Pearson-χ^2^, Fisher exact, and Wilcoxon rank-sum tests, as appropriate. To adjust for the large case mix, univariate and multivariate, unconditional, logistic regression analyses were added. Five to eight predictor variables per outcome variable [22] as well as key variables for collinearity and interaction were analyzed. We used Stata™ software (version 15.0, Stata Corporation, College Station, TX, USA) and considered *p*-values ≤ 0.05 (two-tailed) as significant.

## 3. Results

### 3.1. Study Population

A total of 69 adult patients with “non-diabetic surgical site foot infections”, of whom 61% were biological males (42 patients) and 39% were women (27 patients), were included. Of these, nineteen patients (28%) yielded diabetes mellitus, and eighteen cases (26%) presented with an infected TAA [22]. The median age and BMI were 59 years and 29.4 kg/m^2^, respectively. Other variables, such as the duration of antibiotic use and the number of surgical debridements, were equally distributed among both groups (Table 1).

### 3.2. Diagnosis of Infection

From the intraoperative samples, a total of 52 different bacterial constellations (68% monomicrobial versus 32% polymicrobial infections) were retrieved. The three most common individually detected pathogens were (methicillin-susceptible) *Staphylococcus aureus* in 29% (20 episodes), followed by *S. caprae*/*capitis* and (methicillin-resistant) *S. epidermidis* in 6% of the episodes, whereas the latter two were mainly seen in implant-related infections. Figure 1 displays the number of the most relevant pathogens in quantitative cultures per SSI episode.

### 3.3. Therapy of Infection

Based on international SSI definitions [1,2] as well as the exclusion of recurrent episodes, practically all of our infections were considered “acute” and treated with a combined approach of surgical measures, a nursing approach, professional wound care, internist consultations, and (targeted) systemic antibiotic therapy. The median number of surgical debridements for SSI was 1 (range, 1–4 interventions; IQR, 1–2 surgeries), whereas the number of previous surgeries before SSI onset ranged from 1 to 17. The number of surgeries did not differ significantly between the two groups (Table 1), and 26%, or 18 patients, received additional negative-pressure wound care. There is no hyperbaric oxygen therapy available at our center.

The median duration of systemic antibiotic treatment after the first debridement was 42 days (interquartile range (IQR), 31–76 days; 95% confidence interval (CI), 42–48 days). Upon stratification, 27% of patients received less than 6 weeks of systemic antibiotic therapy, 15% received between 6 and 12 weeks, and 28% received therapy for more than 12 weeks. In the initial years of the cohort (2014 to June 2018), the first 7–14 days of antibiotic therapy was applied parenterally. Later, the length of intravenous application was reduced to 3–5 days [23]. The daily doses were determined according to the upper limits of the Swiss recommendations (www.compendium.ch, accessed on 11 October 2024) and were only reduced in the case of cachexia or renal insufficiency. In the population presented here, no antimicrobial agents in dressings or as intraosseous agents in combination with systemic agents were applied due to the lack of benefit in recent reviews [14] and the acute nature of foot SSIs (by reserving local therapy more frequently for chronic osteomyelitis cases). We noted 135 different antibiotic regimens administered sequentially or concomitantly among the entire study population. Parenteral or oral co-amoxiclav was the most frequent regimen (42% or 27 patients), followed by clindamycin (11%; 7 patients), co-trimoxazole (9%; 6 patients), and vancomycin (6%). Rifampicin combination occurred in six (9%) cases.

Therapy further included different types of immobilization, such as a dorsal splint in a neutral position for 52% (36 patients), no device-related immobilization for 33% (23 patients), and a lower leg cast for 7% (5 patients). The devices were regularly controlled on an outpatient basis in our specialized center. Our healthcare professionals, including surgeons, classified the compliance as sufficiently good. If we had been aware of a major compliance issue in wearing the cast, we would have excluded the patient.

### 3.4. Outcomes of Infection Therapies

A total of 83% (*n* = 57) of all SSI episodes were cured at the first therapeutic approach. Inversely, 17.4%, or 12 patients, sustained “Clinical Failure” after the end of the treatment, of which 13%, or 9 patients, experienced “Microbiological Recurrences” (Table 2) presenting with the same pathogen as in the index infection.

### 3.5. Associations with Clinical Remission and Microbiological Recurrences

We added a univariate and multivariate logistic regression analysis to adjust for the broad case mix (Table 2). Regarding the outcome of “Microbiological Recurrence”, only nine cases with this outcome were present in this study. This paucity excluded a multivariate adjustment. Hence, only univariate results are shown regarding “Microbiological Recurrence”. Concerning “Clinical Remission”, the only (inverse) significant associations were diabetes mellitus (odds ratio 0.12) and the presence of an infected TAA (OR 0.17 in the univariate analysis and 0.10 in the multivariate analysis). Patient characteristics, such as age, sex, BMI, or an infection due to *S. aureus*, did not influence remission. Likewise, the use of negative-pressure wound care or the number of surgical debridements did not alter any outcomes. Importantly, the duration of postsurgical antibiotic treatment had no role, with an odds ratio of 1.0 being associated with a very narrow confidence interval (0.96–1.02) as a continuous variable and similar values being obtained in the stratified analyses. Of note, the goodness-of-fit result was insignificant, and the receiver operating curve (ROC) value yielded 83%, reflecting that our final model has very good accuracy. We are confident that our results are very likely not underpowered.

## 4. Discussion

A total of 83% of our orthopedic foot SSIs were cured at the first therapeutic approach, with one debridement and targeted systemic antibiotic therapy. The duration of postsurgical antibiotic use did not influence “remission” or cause any “Clinical Failure” or a “Microbiological Recurrence” with the same pathogen. Formally extending the antibiotic utilization beyond six weeks did not alter the therapeutic outcome. Indeed, in the multivariate logistic regression analyses, the duration of postsurgical antibiotic use was indifferent when computed as a continuous variable (odds ratio, 1.0; 95% confidence interval, 0.96–1.03) or stratified into three large groups composed of 6-week steps. Similarly, the patient’s age, sex, and BMI; the number of surgical debridements; the use of negative-pressure wound care; or the involvement of *S. aureus* could not alter the outcomes. The only significant associations with more “Clinical Failures” were diabetes mellitus [4] and an infected TAA [23,24], which were both probably associated with wound problems in the aftermath of debridement.

The latter findings are not surprising. The ideal surgical strategy for infected TAA remains unknown [24]. Meticulous debridement, antibiotics, and implant retention (DAIR) in acute TAA infection are considered as acceptable options, without much differentiation between the prosthesis types. DAIR should be performed meticulously, ensuring that all necrotic or infected tissues are removed and modular parts of the prosthesis, if any, exchanged. The infected joint should also be irrigated with antiseptic solutions. Equally concerning TAA infections, these general prerequisites for DAIR are formally not different from those for other prosthetic infections. However, the success of DAIR in TAA is particularly poor [24]. The best evidence is reported by Kessler et al. [24], where the authors investigated 34 cases of TAA infections, out of which 21 were treated with DAIR. Remission using DAIR was only achieved in two-thirds of all cases (14 of 21, 67%), which is a lower success rate than for hip or knee arthroplasty DAIRs. Therefore, DAIR in TAA should be rigorously discussed and should not be considered routinely in each acute SSI. The alternative of arthrodesis or a two-stage exchange (mostly with transient external fixation) might be more suitable in complicated cases.

Our general findings are in line with the literature on non-diabetic foot SSIs [2,8] and similar to our own experience in the field of (ischemic) DFIs [11,13]. Certainly, there is an overestimated effect of the prolongation of antibiotic use among orthopedic surgeons in terms of precautionary benefits. For DFI, a recent review concluded that the duration of systemic antibiotic therapy for osteomyelitis should not exceed 4–6 weeks [11] independently of the underlying surgical approach. This also applies to non-diabetic foot infections. With our retrospective results, we currently target prospective randomized trials aiming to reduce the overall exposure to therapeutic antibiotic agents in infected orthopedic patients. An example is in the SALATIO trials, where we randomize implant-free foot infections between three and six weeks of antibiotic prescription [17] and thus reduce the usual antibiotic therapy period to half. Likewise, in the case of infected implants, the randomization is between six and twelve weeks of antibiotic treatment [17].

Our study has limitations, including its retrospective nature, with some heterogeneity in the patient characteristics, which may lead to possible biases by confounding factors. Likewise, this study was conducted in a single center, a tertiary referral center, which might also hamper the generalizability of our findings. Statements on general compliance (i.e., the correct intake of prescribed antibiotics or adequate immobilization and wound care) in this study cannot be made. For example, one study examining the risk factors for DFI stated that 72% of the affected patients were non-compliant with the prescribed treatment [25], suggesting that non-compliance could be a relevant and underreported risk factor for therapeutic failures. The sample size was small regarding “Microbiological Recurrence”. While we can easily reconstruct the parenteral part during hospitalization, we have no control on patients’ compliance in oral therapy at home. This is a systematic shortcoming of most antibiotic-related trials in an outpatient setting. Likewise, we cannot incorporate single antibiotic agents (or bacterial species) in our multivariate analyses. The variance of detailed variables was too large to do so. Nevertheless, *S. aureus*, as the most common pathogen, failed to alter outcomes when compared to the rest. Lastly, we focused our study on one type of infection in foot surgery (i.e., SSI) in a single center of a resource-rich country to concentrate the results on acute pyogenic infections which make up the bulk of infectious problems in non-diabetic orthopedic foot surgery. We exclude “atypical” pathogens that all require much longer therapies beyond 4 to 6 weeks for pyogenic bacteria. For example, tuberculosis in the foot needs several months of combined antibiotic treatment [26] and may need treatment for more than a year. Fungi and actinomyces are treated for at least three or six months [27]. Finally, the opportunistic *Nocardia* spp. equally requires an antibiotic duration of approximately two months in infections involving the foot and ankle [28].

## 5. Conclusions

In our case–control study, the duration of postsurgical antibiotic therapy does not influence remission or failure risks in the adult population with non-diabetic foot infections. Hence, the mere prolongation of antibiotic therapy will not reverse other shortcomings of the therapeutic approach in this patient population, leaving room for improved antibiotic stewardship in future randomized trials.

## Figures and Tables

**Figure 1 life-14-01509-f001:**
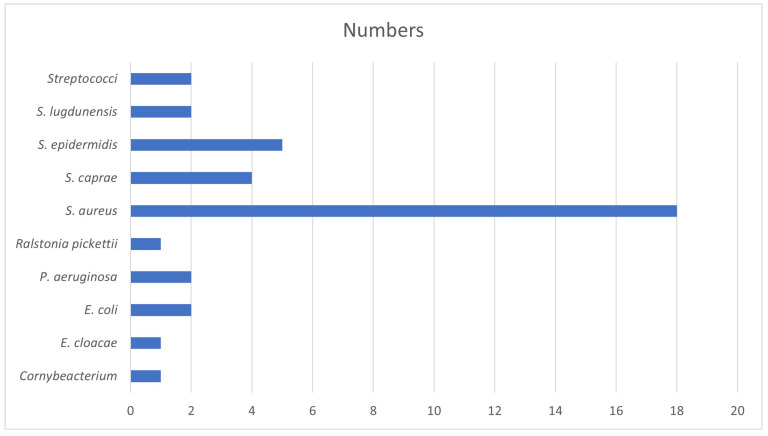
Horizontal *x*-axis: No. of episodes with corresponding predominant pathogen.

**Table 1 life-14-01509-t001:** Patient characteristics in surgical site infections in foot and ankle surgery.

	Overall SSI *n* = 69 (%)	Remission *n* = 57 (82.6%)	Clinical Failure *n* = 12 (17.4%)	*p*-Value
Sex				
Female	27 (39)	25 (44)	2 (17)	
Male	42 (61)	32 (56)	10 (83)	0.079
Median age (years)	59 (49–73.5)	59	61	0.232
Median BMI (kg/m^2^)	29.4 (26–32.3)	29.4	29.4	0.665
Diabetes mellitus	19 (28)	11 (19) *	8 (67) *	** *0.001* **
Total ankle prostheses	18 (26)	11 (19) *	7 (58) *	** *0.005* **
*Staphylococcus aureus*	20 (29)	17 (30)	3 (25)	0.738
Duration antibiotics (days)	42 (range, 31–76)	42 (range, 42–58)	47 (range, 11–120)	0.093
≤6 weeks	27 (39)			
>6 and <12 weeks	14 (20)			
≥12 weeks	28 (41)			
Negative-pressure care	18 (26)	15 (26)	3 (25)	1.000
Surgeries (median number)	1 (1–2)	1	1	0.096

**Footnote**: Data are shown in absolute numbers (% in brackets) or median values (in brackets; range). * Significant *p*-values are printed in ***bold italics.***

**Table 2 life-14-01509-t002:** Case-mix adjustments in logistic regression analyses.

	SSI Remission *n* = 57 (82.6%)		Microbiological Recurrence *n* = 9 (13%)
	Univariate (OR, 95% CI)	Multivariate (OR, 95%)	Univariate (OR, 95% CI)
Sex			
male	0.26 (0.05–1.28)	0.36 (0.03–4.36)	2.50 (0.48–13.06)
Age (years)	0.97 (0.93–1.01)	0.98 (0.92–1.04)	1.02 (0.98–1.07)
Body mass index (kg/m^2^)	0.97 (0.84–1.12)	-	1.01 (0.85–1.19)
Diabetes mellitus	** *0.12 (0.03–0.47* ** *)*	-	4.11 (0.97–17.41)
Total ankle prosthesis	** *0.17 (0.05–0.64)* **	** *0.10 (0.02–0.73)* **	** *4.52 (1.06–19.29)* **
*Staphylococcus aureus*	1.28 (0.31–5.30)	0.57 (0.08–4.18)	0.67 (0.13–3.53)
Duration of antibiotics	1.00 (0.98–1.02)	1.00 (0.96–1.03)	1.00 (0.99–1.02)
≤6 weeks	1 (default)	1 (default)	1 (default)
>6 and <12 weeks	1.04 (0.17–6.54)	1.75 (0.11–27.36)	1.33 (0.20–9.08)
≥12 weeks	0.64 (0.16–2.57)	0.75 (0.01–53.55)	1.33 (0.27–6.61)
Negative-pressure care	1.07 (0.26–4.49)	0.29 (0.03–2.95)	1.50 (0.33–6.75)
Number of debridement	2.43 (0.63–9.40)	2.04 (0.47–8.93)	0.55 (0.15–2.09)

**Footnote**: Data presented as odds ratios with 95% confidence intervals. Significant results are in ***bold and italics***. BMI and diabetes were excluded from multivariate analysis due to substantial interaction.

## Data Availability

We might share anonymized key data upon reasonable scientific request to the corresponding author.

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
