# Peer review of "Therapy for Non-Diabetic Foot Infections After Orthopedic Foot and Ankle Surgery: The Duration of Post-Debridement Antibiotic Treatment Does Not Alter Failure Risk"

_life, 2024, doi:10.3390/life14111509_

Round 1
Reviewer 1 Report
Comments and Suggestions for Authors
The authors present an interesting and useful study in which failures of a combined surgical and antibiotic therapy for surgical site foot infections was retrospectively investigated. I have just a few minor comments.
1." We used data mining from the hospital’s own medical databases and confirmed the presence of SSI by controlling the individual electronic files.” In Switzerland does retrospective analysis of medical records and associated databases require an IRB approval, or exemption? (I would add a sentence to clarify the situation.) What does “controlling the individual electronic files” mean?
22. It would be helpful to briefly define SSI.
33. Section 2: Please define your study groups in termsd of case-control. (It needs to be clearly stated); one sentence will be sufficient.
44. Table 2: under the DM entry you have a * but it is not footnoted.
55. In your study limitations you should mention small sample size (in regard to microbiological recurrence).
Author Response
Reviewer 1
The authors present an interesting and useful study in which failures of a combined surgical and antibiotic therapy for surgical site foot infections was retrospectively investigated. I have just a few minor comments.
Answer: Thank you very much.
1." We used data mining from the hospital’s own medical databases and confirmed the presence of SSI by controlling the individual electronic files.” In Switzerland does retrospective analysis of medical records and associated databases require an IRB approval, or exemption? (I would add a sentence to clarify the situation.)
Answer: There is an IRB approval, which is equally required in Switzerland. We display it at the end of the manuscript in lines 319-325.
What does “controlling the individual electronic files” mean?
Answer: Yes. We reword now. “Controlling” is replaced by “assessing all“ in line 94.
- It would be helpful to briefly define SSI.
Answer: Thank you. We used the classical SSI definitions. This is a common remark of all reviewers. We define it in the “Materials and Methods” Section in lines 132-139.
- Section 2: Please define your study groups in terms of case-control. (It needs to be clearly stated); one sentence will be sufficient.
Answer: Yes, this is another common remark of the Reviewers. We define it briefly in the part “Statistical analyses” in lines 145-146.
- Table 2: under the DM entry you have a * but it is not footnoted.
Answer: Thank you very much. A typo. We now removed the *.
- In your study limitations you should mention small sample size (in regard to microbiological recurrence).
Answer: Agreed. This is important. We add this shortcoming into the line 284.
Reviewer 2 Report
Comments and Suggestions for Authors
Please find below my comments on the manuscript:
-Abstract: the type of study, study period and the name of the hospital where the study was held are missing. Please add.
-Methods: Please define remission after minimal follow-up. Did it include microbiological eradication and improvement in clinical/hematochemical inflammatory markers?
-Also in methods add which methods were used to detect pathogens for e.g VITEK 2 etc, and which criteria were used to evaluate the resistance profiles of these pathogens: EUCAST OR CLSI?
-Results, lines 147-148: “In the early years of the cohort (2014 to June 2018), the initial 7-14 days 147 of antibiotic treatment occurred parenterally. Afterwards, the initial intravenous part was 148 reduced to 3-5 days [23].” Could the authors include in the analysis not only the total duration of treatments but also the duration of intravenous and oral ones?
Author Response
Reviewer 2
Please find below my comments on the manuscript:
-Abstract: the type of study, study period and the name of the hospital where the study was held are missing. Please add.
Answer: We agree and add this information to lines 15-16.
-Methods: Please define remission after minimal follow-up. Did it include microbiological eradication and improvement in clinical/hematochemical inflammatory markers?
Answer: We already state that “Remission was a complete absence of any clinical, laboratory, and/or radiological evidence of a new or persistent problem” (lines 126-127). However, we understand Reviewer 2 and now complete by saying that we did not routinely rely on laboratory control parameters. as their use is not evidence-based in orthopedic foot surgery (lines 128-130). For this, we add two new references 19, 20.
-Also, in methods add which methods were used to detect pathogens for e.g. VITEK 2 etc, and which criteria were used to evaluate the resistance profiles of these pathogens: EUCAST OR CLSI?
Answer: VITEK 2 and EUCAST criteria for Switzerland. We mention it very briefly in lines 140-142. Our laboratory used “routine” methods.
-Results, lines 147-148: “In the early years of the cohort (2014 to June 2018), the initial 7-14 days 147 of antibiotic treatment occurred parenterally. Afterwards, the initial intravenous part was 148 reduced to 3-5 days [23].” Could the authors include in the analysis not only the total duration of treatments but also the duration of intravenous and oral ones?
Answer: In contrast to intravenous treatment, the bulk of oral antibiotic treatment occurred in the outpatient setting. We know how long we prescribed, but not how long the patient really took his/her antimicrobial medication, and if the corresponding compliance was 100%. We discuss these aspects in the “Limitation” (lines 284-288).
Moreover, please bear in mind that the distinction between intravenous vs. oral therapy is not sharp. A patient can have IV medication together with oral rifampin in implant-related infections, or a mix of forms in polymicrobial infections. All these aspects make the statistical analysis, as demanded, quasi impossible.
Reviewer 3 Report
Comments and Suggestions for Authors
Major points
- The Introduction should also contain an extended paragraph regarding the most common pathogens in both DFI and SSI; are there differences in pathogen frequency between these two conditions?
- Line 73 – The criteria of defining SSI should be added to the manuscript
- Line 90 – How well-regulated diabetes mellitus defined? While some precautions could have been taken here, I believe it would have been “safer” from a methodological standpoint to exclude all diabetic patients
- Line 128 – 132 - From a microbiological standpoint, the bacterial constellations provide great interest. This section should be expanded and the most common pathogens for both mono- and polymicrobial infections should also be presented in a separate Table
- Discussion – Does local wound therapy with antimicrobials or antiseptics not provide any benefits in the case of SSI/ DFI? While the results are interesting and point towards the common trend of improving antibiotic stewardship and shorten treatment durations, I believe that the local treatment deserves a further discussion in regards to advantages/disadvantages
Minor points
- I believe it would be helpful to add the fact that out of the 12 clinical failures 9 were classified as microbiological recurrences in the description of table 1
Author Response
Reviewer 3
Major points
- The Introduction should also contain an extended paragraph regarding the most common pathogens in both DFI and SSI; are there differences in pathogen frequency between these two conditions?
Answer: We mostly agree. We now add the required information into the chapter “Introduction” (lines 44-63). However, and very importantly, our article is not about DFI or community-acquired foot infections. The microbiology of foot infections can be very diverse depending on the nosocomial origin, underlying implants and ulcerations, the geographical regions, and associations with other medical pathologies such as ischemia or skin maceration. We do comply with the demand of Reviewer 3 but cannot review the large epidemiology of foot infections all over the world and settings.
- Line 73 – The criteria of defining SSI should be added to the manuscript
Answer: Thank you. We used the classical SSI definitions. This is a common remark of all reviewers. We define it now in “Materials and Methods” in lines 132.139,
- Line 90 – How well-regulated diabetes mellitus defined? While some precautions could have been taken here, I believe it would have been “safer” from a methodological standpoint to exclude all diabetic patients.
Answer: Definition was based on clinical grounds. We are a surgical clinic performing a retrospective analysis on surgical files. Only in prospective studies conducted by endocrinologists you can include or exclude according to strict laboratory values. We mention this briefly in lines 115-116.
Secondly, diabetes mellitus is a well-known risk for the occurrence of SSI. In contrast, the presence of diabetes is not necessarily associated with failures in localized orthopedic infections, in contrast for example for systemic sepsis in the ICU. Even in diabetic foot infections, failures are mostly due to shortcomings of off-loading, ischemia and compliance, and not primarily because of bad glycemic variables.
With your permission, we do not want to complicate our analyses by stratifying our small sample size by creating different groups for a clinical variable (diabetes mellitus) that is more related to failures through ischemia and wound breakdowns than with antibiotic treatments.
- Line 128 – 132 - From a microbiological standpoint, the bacterial constellations provide great interest. This section should be expanded and the most common pathogens for both mono- and polymicrobial infections should also be presented in a separate Table.
Answer: Ok. Thank you. We already add more information (and references) to the “Introduction” (lines 44-63) and display our pathogens as a new Figure 1 (not Table).
- Discussion – Does local wound therapy with antimicrobials or antiseptics not provide any benefits in the case of SSI/ DFI? While the results are interesting and point towards the common trend of improving antibiotic stewardship and shorten treatment durations, I believe that the local treatment deserves a further discussion in regards to advantages/disadvantages
Answer: Yes and no. Yes, if local antimicrobials (topical or intraosseous) are given solely, or maybe compared to wound debridement alone. No, if local are combined with a targeted systemic antibiotic agent and the wounds are regularly debrided. In these combined episodes, almost all retrospective studies failed to show a superiority of add-adding. We already mention this with our own reference 14 and now also expand our explanations 195-198.
Minor points
- I believe it would be helpful to add the fact that out of the 12 clinical failures 9 were classified as microbiological recurrences in the description of table 1
Answer: Yes. It is above all the small sample size for that specific outcome. We now mention it, in one sentence, in the” Limitation” (line 284).
Reviewer 4 Report
Comments and Suggestions for Authors
Please, see the atached file.

Minor editing is needed.
Author Response
Reviewer 4
The possibility of shortening the duration of antibiotic therapy for non-diabetic foot infections is an important clinical issue and can significantly affect the prevention of the development of antimicrobial resistance. Therefore, this manuscript is significant from a clinical and epidemiological point of view.
Answer: Thank you very much.
However, in my opinion, the work needs to be improved before publication. First of all, too many self-citations were observed. The mentioned references should be replaced by relevant references of other authors.
Answer: Ok. Together with the demands of other Reviewers for more explanatory text, we now increase the number of references by eight (among which only one of ours) and reduce the number of our old references by three. Many of our own references are necessary for lecture. It is no coincidence that our group wrote this specific paper.
Furthermore, the methodology is insufficient. The authors state that this is a case-control study, and it is not clear what is a case and what is a control.
Answer: Yes, this is another common remark of the Reviewers. We define it briefly in “Statistical analyses” in lines 144-147.
Also, there is a lack of detailed data on all isolated bacteria, on all antibiotics that the patients received; it should be presented in detail, at least as supplementary material. I think the authors should do a detailed analysis of all the available data, especially indicate which infections (which bacteria) were treated with which antibiotic and for how long.
Answer: This is impossible and would require a general sample size of several hundred of SSI episodes. For our study population, it would represent pseudo-science. We already state that we noted 52 different microbial constellations (line 164) and 135 various antibiotic regimens (line 199). The latter was, moreover, sometimes sequentially, and sometimes concomitantly administered. With only few failures as the primary outcome variable, we cannot afford such analyses. Instead, we picked the most important pathogen, S. aureus, and analyzed its infections separately (Tables 1 and 2). S. aureus did not alter outcomes when compared to the rest of pathogens.
Furthermore, patients who developed distant complications should be included in the study.
Answer: Our article is about local orthopedic foot SSI. This treatment is different if the patient has, let’s say, a concomitant primary or secondary endocarditis. Likewise, we also lacked patients with other concomitant infections due to different pathogens, such as pyelonephritis. In any case, such additional analyses would be beyond the scope of this study. In contrast, these difficulties are the exact reason why we focused on SSI; to contain the case-mix and avoid the mix of therapies in our clinical and elsewhere. We make this clearer in the “Limitations” (lines 291-294).
The work would be much more valuable if the patients were divided into two groups: those with local infection and patients with distant complications.
Answer: Concerning the distant infections, please see above. Furthermore, a separate stratum with foot infections PLUS the distant other infections is statistically practically impossible because of data paucity and the heterogeneity of the remote infections themselves. We are unaware of any scientific publication in Infectious Diseases that we have performed such a complicated analysis. All streamline to the most frequent groups of interest (which we did to the best possible).
Also, they should not exclude patients with infections caused by, as the authors state, "atypical pathogens". All these data are very important and can significantly affect survival and the main conclusion of this study, which is the length of antibiotic treatment.
Answer: Yes. We have to be more precise. We have included skin commensals and virulent pathogens such as Pseudomonas and streptococci. However, what we mean by atypical pathogens are for example tuberculosis, fungi, actinomyces; Nocardia, etc., that require per se a much longer treatment. We cannot mix pyogenic bacteria and tuberculosis. No one does. We clarify this better (lines 295-300) and add three new corresponding references (references 26-28; lines 297-300).
Round 2
Reviewer 4 Report
Comments and Suggestions for Authors
Line 62: P. aeruginosa (italic)
Line 139: please add reference for used EUCAST protocol
Figure 1: all latin bacterial names should be written italic (S. lungdunensis...)
Line 274: distance between " the prescribed"
Line 276: delete "iv" and "v"
I would like to thank the authors for their answers and modifications that were made on the manuscript which is, in my opinion, significantly improved.
Only a few minor changes are needed:
Line 277-278: Please delete or rephrase this sentence (I can not belive that physician doesn't know which antibiotic was given to patient!?)
Line 280: please delete "vi"
Line 284: delete "vii"
Author Response
Line 62: P. aeruginosa (italic)
Answer: Done.
Line 139: please add reference for used EUCAST protocol.
Answer: Yes. This is now the new reference 22 (line 139). We reference their official “standard operation procedures” which they make public. Hence, this reference is their website. The microbiologists do not reference a publication by themselves.
Figure 1: all Latin bacterial names should be written italic (S. lugdunensis...).
Answer: Done. The first word is “Streptococci”, which we also made in italic to maintain harmony in the Figure.
Line 274: distance between " the prescribed".
Answer: Now separated.
Line 276: delete "iv" and "v"
Answer: Done. We remove now all the Latin / Roman numerations in the “Limitation” chapter.
I would like to thank the authors for their answers and modifications that were made on the manuscript which is, in my opinion, significantly improved.
Answer: Thank you very much.
Only a few minor changes are needed:
Line 277-278: Please delete or rephrase this sentence (I cannot believe that physician doesn't know which antibiotic was given to patient!?)
Answer: Done. We deleted without replacement.
Line 280: please delete "vi"
Answer: Done.
Line 284: delete "vii"
Answer: Done.
